

# 1 Regional-scale landslide risk assessment in Central-Asia

Francesco Caleca[1*], Chiara Scaini[2], William Frodella[1], Veronica Tofani[1]
1 University of Florence, Department of Earth Sciences, via G. la Pira 4, 50121 Florence, Italy.
2 National Institute of Oceanography and Applied Geophysics – OGS, Borgo Grotta Gigante , Sgonico (Trieste),
Italy.
*Correspondence to: Francesco Caleca (francesco.caleca@unifi.it)
**Abstract.**
Landslides are widespread phenomenon that occur in any terrestrial area with slopes, causing massive property damage and,
in the worst-case scenario, human losses. This propension to suffer losses is particularly high for developing countries due to
their urban development, population growth and drastic land use changes. Social and economic consequences of landslides
can be reduced through detailed planning and management strategies, which can be aided by risk analysis. In this study, we
performed a detailed quantitative risk analysis for landslides in the whole Central-Asia (4,000,000 km$^2$). Landslide-induced
risk was computed in terms of exposed population and expected economic losses to buildings and linear infrastructures
(roads and railways) adopting a 200 m spatial resolution. The purpose of our study is to produce the first regional-scale
landslide risk assessment for Central-Asia in order to inform regional-scale risk mitigation strategies and it represents an
advance step in the landslide risk analysis for extremely broad areas.

## 19 1 Introduction

Landslides are widespread phenomena that occur in any terrestrial area with slopes and cause huge damages to properties
and in the worst case, they are responsible of human losses (Petley 2012). Landslide events can be triggered by many
different factors, the main causes recognized by the geoscience community are attributable to tectonic, climatic (e.g. intense
rainfall) and human (e.g. construction, mining) activities (Petley et al. 2007; Huang et al. 2012; Froude and Petley 2018;
Segoni et al. 2018; Turner 2018). However, the increasing occurrence of extreme events and their effects related to climate
change certainly represent a further factor in the propensity of slopes to instability (Gariano and Guzzetti 2016; Haque et al.
2019). Every year, significant loss of lives and economic damages are caused by landslides over the whole globe; according
to Haque et al. (2019) landslides should be ranked as the 4th biggest killer globally among natural disasters since yearly they
cause more than 4000 direct life losses and over 7800 indirect (due to landslides triggered by other natural hazards).
Similarly, the urban development in risk-prone locations, land use changes, environmental degradation and weak planning
strategies are responsible of the severe economic losses due to landslides.





Therefore, social and economic consequences of landslides can be reduced by means of detailed planning and management
strategies, which can be facilitated by risk analysis in order to make rational decisions on the allocation of funds to plan
mitigation measures (Dai et al. 2002).
Risk is defined as the measure of the probability and severity of an adverse effect to life, health, property or the environment,
while risk analysis is the use of available information to estimate the risk to exposed elements from hazards (Fell et al.
2005). According to the existing literature, risk analysis can be performed in two different ways: qualitatively or
quantitatively. Qualitative analysis report risk using word form, descriptive or numeric rating scales (e.g low, moderate and
high) to describe the magnitude of potential consequences and the likelihood that those consequences will occur     (Abella
and Van Westen 2007; Wang et al. 2013). Contrarily, quantitative risk analysis is based on numerical values of the
probability, vulnerability and consequences, and resulting in a numerical value of the risk applying the equation proposed by
Varnes and IAEG Commission on Landslides (1984): R(I) = H x V(I) x E, where R is landslide risk, H is the landslide
hazard, V is the vulnerability of the exposed elements, I is the intensity of landslide and E the value of elements at risk. In
accordance with Corominas et al. (2014), quantitative risk analysis (QRA) allows risk to be quantified in an objective and
reproducible manner comparable from one location to another. The general framework of QRA includes different steps:
hazard identification and assessment, location of elements at risk and their relative exposure, vulnerability assessment and
risk estimation (Dai et al. 2002; Fell et al. 2008; Corominas et al. 2014).
Landslide hazard assessment aims to identify which areas are most prone to trigger landslides with a certain intensity within
a given period of time (Guzzetti et al. 2005; van Westen et al. 2006; Corominas et al. 2014; Lari et al. 2014). Therefore,
landslide hazard evaluation is carried out by means of the analysis of three different probabilities: probability of landslide
size, temporal probability of landslides and spatial probability of landslides also known as landslide susceptibility. This latter
is the likelihood of a landslide occurring in an area on the basis of the local terrain conditions (Brabb 1984; Kanungo et al.
2012; Reichenbach et al. 2018) and it is the initial step towards landslide hazard, but it can be also considered as a final
product (Corominas et al. 2014). In particular, in the case of lack of available data related to the landslide frequency and size,
landslide hazard can be approximate to the landslide susceptibility (Caleca et al. 2022).
Vulnerability plays an important role to define the consequences of a landslide event and it refers to the degree of loss of a
given element at risk, vulnerability is generally expressed on a scale of 0 (no loss) to 1 (total loss) (Glade 2003; Uzielli et al.
2008; Li et al. 2010; Corominas et al. 2014; Peduto et al. 2017). Vulnerability assessment is related and performed on the
basis of landslide intensity and magnitude, nevertheless for risk analysis referred to very vast study areas and for which it is
very complicated to retrieve homogenous data to estimate it, vulnerability can be assumed equal to total damage (e.g total
loss) (Glade 2003).
Exposure analysis is an intermediate stage of risk assessment linking the susceptibility and hazard assessment with the value
of elements at risk (Pellicani et al. 2014). According to the literature, exposure is an attribute of considered elements at risk
that are potentially affected by a landslide (Lee and Jones 2004; Corominas et al. 2014). In the case of population, it is
generally expressed as the number of people exposed to hazardous phenomena, and further distinction can be made based on



demographics or socio-economic indicators (Maes et al. 2017). As for the physical exposed assets (e.g. buildings,
transportation and other infrastructures), exposure is quantified by the economic value of the elements (Schuster and
Fleming 1986; Schuster and Turner 1996). Exposure assessment methods strongly rely on the spatial scale and can be carried
out at global or regional-scale (Emberson et al. 2020; Pittore et al. 2020) with the necessary assumptions and simplifications
(e.g. spatial aggregation). However, exposure assessment can also be developed at the local-scale and for single assets
(Garcia et al. 2016). Commonly, one of the financial risk metrics is the reconstruction cost, i.e. the amount of money needed
to reconstruct the asset following the current regulations (Benson and Clay 2004). In recent times, an increasing number of
datasets (e.g. high-resolution population and land-use data, remote sensing products) supports the assessment of damage and
risks in a timely manner. However, characterizing exposed assets for the purpose of disaster risk assessment is still one of the
pushing challenges of current disaster risk reduction agenda (Kreibich et al. 2022).
In the last two decades, several studies dealing with QRA have been proposed, however it is worth nothing that the majority
of performed analysis have been limited to test sites or basin scale at most (Ko et al. 2003; Catani et al. 2005; Michael-Leiba
et al. 2005; Remondo et al. 2005, 2008; Zêzere et al. 2008; Jaiswal et al. 2011; Lu et al. 2014; Uzielli et al. 2015; Corominas
et al. 2019; Jinsong Huang et al. 2020; Ferlisi et al. 2021; Caleca et al. 2022). Nevertheless, when the case study is
represented by very broad areas (e.g nations), QRA is very difficult to perform due to the difficulty to obtain homogeneous
and complete hazard and exposure datasets. Most  studies rely on the definition of indicators that are an oversimplification of
the QRA framework, but very easy to understand and update (Abella and Van Westen 2007; Puissant et al. 2014; Guillard-
Gonçalves et al. 2015; de Almeida et al. 2016; Trigila et al. 2018; Bezerra et al. 2020; Pereira et al. 2020; Segoni and Caleca
83  2021).

The purpose of this paper was to perform a detailed landslide QRA for a very broad area, which is represented by the whole
region of Central-Asia. Despite the documented damages due to landslides in the past, to our knowledge there is no regional-
scale landslide risk assessment available for Central Asia.  In addition, landslide-induced risk in the region is expected to
increase due to urban development, population increase and land use changes. In this study, we produce the first regional-
scale landslide risk assessment in order to inform regional-scale risk mitigation strategies. Landslide-induced risk was
computed in terms of exposed population and expected economic losses to buildings and linear infrastructures (roads and
railways); obviously since the selected case study is very vast, some approximations within the framework of risk analysis
have been implemented. The final goal of this work is to identify in which areas highest losses could occur in order to
provide a very useful tool for possible mitigation measures and land-planning policies.
**2 Study area**
The region of Central-Asia is constituted by the following countries: Kazakhstan, Turkmenistan, Uzbekistan, Tajikistan and
Kyrgyz Republic (Fig.1) and it covers an area of about 4,000,000 km$^2$. From a geographical point of view, Central-Asia
shows a varied geography including mountain chains, grassy steppes and vast deserts (Kyzyl Kum, Taklamakan). The





southern and eastern sectors of the region are mountains areas, mainly covered by the Tien Shan chain with summits higher
than 7000 m (Charreau et al. 2006; Strom 2010). The geological history of Tien Shan range is very complex and it is
characterized by a Palaeozoic subduction process (Burtman 1975; Windley et al. 1990) and after by a new Cenozoic phase,
consequent to a tectonic activity due to the convergence between India and Eurasia (Molnar and Tapponnier 1975; Davy and
Cobbold 1988; Havenith et al. 2006; Buslov et al. 2007). Tien Shan consists of E-W mountains ridges marked by several
fault systems, the most important of those is the Talass-Fergana Fault Zone, which divides the western Tien Shan from the
central one (Trifonov et al. 1992).
The most common landslide events in Central-Asia are rockslides/rock avalanches, rotational/translational slides and
mud/debris flows and they are mainly caused by earthquakes, floods, snowmelt and intense rainfall (Kalmetieva et al. 2009;
Behling et al. 2014; Golovko et al. 2015; Havenith et al. 2015; Saponaro et al. 2015; Strom and Abdrakhmatov 2017, 2018).
Landslides seismically triggered are very common and most of the large mapped ones were caused by high-magnitude
earthquakes, even prehistoric, associated with extreme climate events like intense rainfall or snowmelt (Havenith et al. 2003,
2015; Strom 2010; Strom and Abdrakhmatov 2018; Piroton et al. 2020). At the regional scale, Tajikistan and Kyrgyz
Republic are the countries most impacted by landslide due to their geological and geomorphological settings; about 50000
landslide have been mapped in Tajikistan (Thurman 2011), while Kyrgyz Republic has been affected by 5000 landslides.
Emberson et al. (2020) show that the population fraction exposed to landslides in Central Asia exceeds the 10% and 20% in
Tajikistan and Kyrgyz Republic respectively. However, other sectors that are not located in the above-mentioned countries
(e.g the Almaty region in Kazakhstan or the Tashkent one in Uzbekistan) are also affected by landslide phenomena, mainly
due to the increase of the anthropic pressure and activities, which certainly rise the number of elements at risk potentially
interested and therefore the level of exposure in the study area.


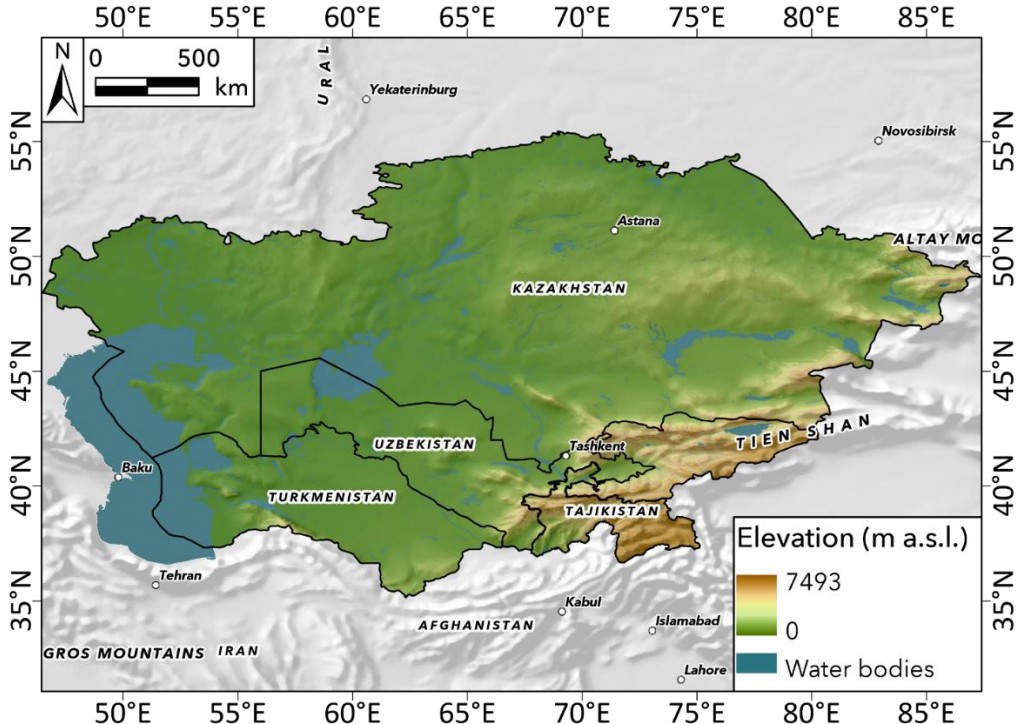

Fig.1 Location and elevation of the study area.

## 3 Data and methods

In this paper landslide risk was evaluated applying the well-known risk equation proposed by Varnes and IAEG commission on landslide (1984), where risk (R) is defined as the multiplication of three parameters: hazard (H), vulnerability (V) and exposure (E). Nevertheless, since the study area is characterized by a huge areal extension, some approximations within the risk analysis were performed to fix the heterogeneity and the lack of data to assess the different landslide risk parameters, specifically simplifications were applied into the landslide hazard and vulnerability assessment. The hazard component was considered as the spatial probability occurrence of landslides (susceptibility) in the study area since it was impossible to retrieve suitable information to evaluate the temporal and landslides size probabilities from the available databases. Besides, vulnerability was set equal to 1, or rather the maximum possible degree of loss, due to the lack of data necessary to assess separately the physical vulnerability of each exposed elements. Regarding exposure component, we employed a very-recently and detailed database developing during the EU-Funded Strengthening Financial Resilience and Accelerating Risk Reduction (SFRARR) program. The research program, implemented by World Bank and the Global Facility for Disaster Reduction and Recovery (GFDRR) was implemented between 2020 and 2022 of assets exposed to flood, earthquake and landslides for Central Asia. The exposure dataset (Scaini et al., submitted-A, Scaini et al., submitted-B) was produced at a resolution of 100 (population) and 500m (buildings) to support regional-scale risk assessment. However, for the purpose of



landslide risk assessment, the spatial resolution of the buildings' layer should be increased to grap the spatial distribution of
exposed assets and avoid risk overestimation. Further details on how the layers were developed in the context of landslide
risk assessment are provided in section 3.1 and 3.2. Landslide risk was computed by estimating the number of exposed
population and the expected monetary losses to different types of buildings and transportation systems. The calculation was
performed at 200 m spatial resolution discarding flat areas (slope lower than 5 degrees) where landslides are not expected as
a geomorphological process. Risk is then expressed in monetary terms (i.e. United States Dollars, USD), as expected
economic losses across the study area.

| Input data | Risk parameter | Resolution | Reference |
|---|---|---|---|
| SRTM DEM | Grid analysis | 90 m | Farr and Kobrick (2000) |
| Landslide susceptibility map | Hazard | 70 m | Rosi et al. (2023) |
| Spatial distribution of population | Exposure | 100 m | Scaini et al.(submitted-A) |
| Spatial distribution of residential buildings and relative reconstruction costs. | Exposure | 500 m | Scaini et al.(submitted-A) |
| Spatial distribution of commercial buildings and relative reconstruction costs | Exposure | 500 m | Scaini et al.(submitted-B) |
| Spatial distribution of transportation systems and relative reconstruction costs | Exposure | variable | Scaini et al.(submitted-B) |

**Table 1. Input data**
**3.1 Landslide hazard**
The hazard component of risk was considered the spatial probability of landslides occurrence; we are aware that this
procedure represents a simplification within the QRA framework. Nevertheless, according to Corominas et al. (2014)
landslide susceptibility can be considered as a final product, especially in small scales analyses or in studies where
information to estimate both temporal probability of occurrence and size one about landslides are insufficient (Caleca et al.
2022). Therefore, the hazard assessment in the present study relies on already published landslide susceptibility map of



Central-Asia (Rosi et al. 2023). The map was obtained applying a machine learning algorithm, the Random Forest
Treebagger (Breiman 2001; Brenning 2005), which application in landslide susceptibility studies is well-consolidated
(Catani et al. 2013; Trigila et al. 2013; Youssef et al. 2016; Lagomarsino et al. 2017; Taalab et al. 2018; Kavzoglu et al.
2019; Merghadi et al. 2020). The landslide susceptibility map was obtained implementing the algorithm over the whole
study area, instead of processing each single country; 26 different predictors (e.g lithology, distance from faults, Peak
Ground Acceleration maps, maps related to precipitation) were employed in the model optimization and training. The
algorithm was set to work in classification mode identifying presence or absence of landslides (dependent variable) and then
for each pixel the probability to be classified as landslide was evaluated. The accuracy of model performance was evaluated
by means of the AUC (area under the receiver operator characteristic curve), which mean value was equal to 0.93, showing
an extremely excellent result for susceptibility modelling.
The original landslide susceptibility map, based on a 70 m spatial resolution, was upscaled to the selected resolution of this
work (200 m), the values of probability of landslide occurrence were averaged over each 200 m cell of the reference grid
used for risk analysis, providing a spatial hazard index ranging from 0 to 1.
It is worth noting that the input susceptibility map is not related to a specific type of landslide since the adopted landslides
inventories to train the model did not report the typology of the event, therefore the performed risk analysis does not refer to
a specific type of landslide phenomena as well.

**3.2 Exposure**

The exposure assessment proposed in this paper was carried out separately for the following elements at risk: buildings,
transportation systems and population. Concerning buildings and transportation systems, exposure was evaluated as their
reconstruction cost expressed in United States dollar (USD), while population exposure was expressed in number of lives.

**3.2.1 Population exposure**

The population dataset was developed based on the most recent high-resolution global-scale dataset (Facebook, available at
https://data.humdata.org/organization/facebook at 20-m resolution) complemented with national census data collected for
each of the five Central Asian countries in cooperation with local representatives (Scaini et al., submitted-A) The resulting
exposure layer provides the spatial distribution of population (including gender and age classes) over the whole study area at
a 100m resolution. The population exposure is represented here by the total number of inhabitants in each cell, without
gender and age distinction.

**3.2.2 Building exposure**

In the present study, two different categories of buildings were analysed within the exposure and risk analysis: residential
and commercial. Information about residential buildings were provided by a recent work performed on their exposure and
spatial distribution over the whole study area (Table 1). The regional-scale buildings exposure dataset was based on the




residential buildings exposure model developed by Pittore et al. (2020), which was refined using national-scale data (e.g.
national building census and reconstruction costs). The result is a new exposure dataset which comprises both residential and
non-residential buildings and their economic value on a constant-resolution grid of 500 meters. The resolution of the input
regional-scale dataset was increased to 200m by means of a spatial analysis procedure (Fig.2). First, for each 500-m cell a
mean economic value per building was defined, then the number of buildings was spatially distributed (spatial
disaggregation) employing as proxy the 100-m population grid (Table 1). Then, the reconstruction costs in each 100-m cell
have been obtained multiplying the mean value and the new spatial distribution of residential buildings. finally, the 100-m
resolution exposure value is aggregated by summing the values of each 100m cell to be comparable with the 200 m
landslides susceptibility grid (3.1) used for the analysis. Increasing the resolution of exposure data from 500 to 200m allows
a better spatial representation of exposure and prevents risk overestimation when dealing with local phenomena such as
landslides.

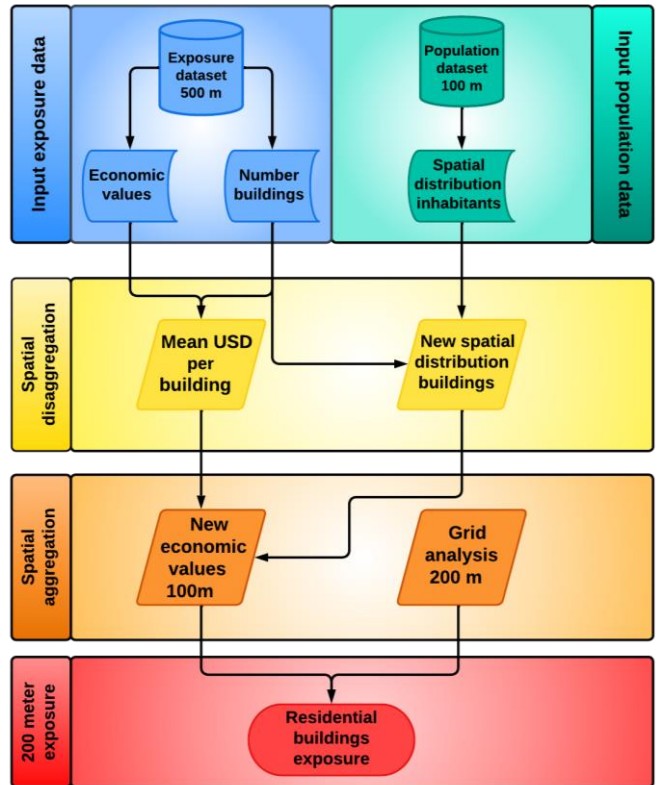

**Fig.2 Flowchart of the disaggregation procedure which distributes the buildings exposed value on the analysis grid at 200-m**
**resolution.**

Exposure of commercial buildings was estimated by means of the commercial building exposure dataset at a 500 m spatial
resolution (Table 1). The layer, developed by Scaini et al. (submitted-B), distinguishes between two commercial buildings





categories: wholesale and services (associated to large buildings) and retail (associated to medium/small business). Besides,
for each typology the number of structures and their relative reconstruction costs were defined. Differently from residential
buildings, commercial buildings were not distributed on a 100-m grid using population as a proxy. This is because
commercial buildings can be located both in populated and non-populated areas. The economic value of commercial
buildings was equally distributed from the original (500 m) to the target (200m) spatial resolution.

**3.2.3 Transportation systems exposure**

The input transportation exposure dataset was developed on the basis Open Street Map data and country-based information
on the length, type and reconstruction cost of each road/railway type (Scaini et al., submitted-B). Here, for the purpose of
landslide risk assessment, we consider two main classes of transportation systems: roads and railways. Specifically, the
exposure layer provided the total length and reconstruction costs of different sub-classes of roads (primary, secondary,
tertiary, motorway and trunk) and railways (conventional and high-speed). The total reconstruction cost is defined for each
linear infrastructure sub-type by multiplying its length and reconstruction cost (USD/m) within each cell.

**3.3 Landslide risk**

Landslide risk has been computed through a quantitative assessment by assessing the probability of expected losses for the
selected elements at risk. The computation is performed on a 200-m grid and only for cells where the landslide susceptibility
is not null. Probability is then classified using a continuous scale ranging from the minimum to the maximum value of losses.
In particular, losses are intended here as the sum of the value of each asset at stake, assuming a vulnerability of 1 (Coriminas
et al.2014). Equally to exposure assessment, risk analysis was performed separately for the selected exposed elements,
producing several specific risk datasets and these results were then combined into a map of total risk. The total risk map was
obtained combining exposure in terms of monetary value. For this reason, the assessment of risk for population was not
included in this computation and it was analysed separately. In this work four different specific risk have been analysed:
population risk. buildings risk; roads risk and railways risk.
Population risk has been computed:

$$R_p = H \times P \qquad \textbf{eq.1}$$

where $R_p$ is the number of lives potentially at risk, H is hazard and P is the mean number of inhabitants within each cell of
the grid analysis.
Buildings risk has been computed:

$$R_b = H \times (E_r + E_c) \qquad \textbf{eq.2}$$

where $R_b$ is the expected loss to buildings, H is hazard, $E_r$ and $E_c$ are the exposure of residential and commercial buildings
respectively.
Roads risk has been computed:





$$R_{ro} = H \times (E_p + E_s + E_t + E_m + E_{tr}) \qquad \textbf{eq.3}$$

where $R_{ro}$ is the expected loss to roads, H is hazard, $E_p$, $E_s$, $E_t$, $E_m$ and $E_{tr}$ are the exposure of primary roads, secondary roads,
tertiary roads, motorways and trunks respectively.
Railways risk has been computed:
$$R_{ra} = H \times (E_{co} + E_h) \qquad \textbf{eq.4}$$

where $R_{ra}$ is the expected loss to railways, H is hazard, $E_{co}$ and $E_h$ are the exposure of conventional and high-speed railways
respectively.
Total risk is the sum of the specific risks of buildings, roads and railways:
$$R_{tot} = R_b + R_{ro} + R_{ra} \qquad \textbf{eq.5}$$


## 239 **4 Results and discussion**

### 240 **4.1 Landslide hazard**

The landslide hazard map of Central-Asia is showed in Figure 3, since most of the study area is constituted by flat areas the
majority of hazard values lies in an interval that can be classified as low-moderate probability occurrence according to
literature overview. In detail, the mean hazard value is 0.37 and about 24% of the analysed area presents hazard values less
or equal than 0.25 and they are mostly located in the northern and western part of Central-Asia. However, there are sectors of
the case study reporting very high values of landslide hazard: the 0.65% of whole Central-Asia showed hazard values greater
or equal than 0.75, that can be classified as very-high probability of occurrence of landslide. The 74% of these cells reported
the maximum value of hazard (1) and most of them are located within the country of Tajikistan and Kyrgyz Republic, that
are mostly covered by the Tien Shan range, which due to its geological and geomorphological settings is very prone to
trigger landslide phenomena. Nevertheless, even several cells of Uzbekistan and Kazakhstan exactly located in the Tashkent
and Almaty regions, present hazard values close or equal to 1.
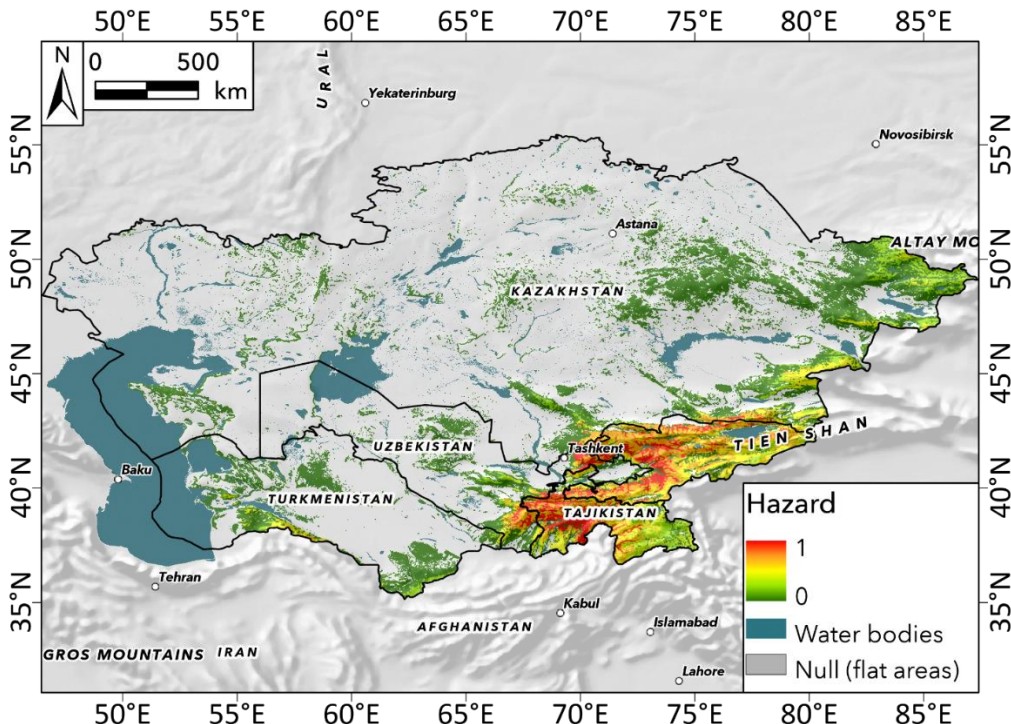

**Fig.3 Landslide hazard map of Central-Asia. Note that flat areas are excluded because they are places not prone to trigger landslides.**

## 4.2 Exposure

The population exposed to landslides is reported in Fig.4A, which shows the total number of inhabitants per cell. Exposed population ranges from 0 to a maximum of 433.97 inhabitants, which is located in the city of Ghafurov, Sughd region in Tajikistan. All population exposed to possible landslide events is located within 1.1% of the cells, with a mean density of 5.7 inhabitants per cell. All the other areas are not inhabited. This is because highly populated areas are not included in the exposure layer since they are sited in floodplains, which are filtered off from the computation because they are not prone to landslides.

Fig.4B shows the spatial distribution of buildings exposure over Central-Asia, obtained by combining the total reconstruction cost of residential and commercial buildings. The total buildings exposure ranges from 0 to 1.39 million USD per cell (corresponding to approximately 35 million USD /Km$^2$), the highest value being located in the city of Almaty (Kazakhstan), at the foot of the Tien-Shan chain. The 0.81% of the analysed area reports a buildings exposure greater than 0, the mean value is approximately 45,000.00 USD per cell and the sum is about 517 million USD.

Note that flat areas, where buildings exposure is higher, were excluded from the risk analysis. The total exposed value of commercial buildings in landslides-prone areas is of 280 million USD, which is greater than residential one. Only the 0.10% of landslide-prone cells have a not-null commercial exposure and the mean exposed value is about 39,000.00 USD.





The total exposure of roads in Central-Asia (Fig. 4C) has been computed summing the exposure of the different road types
(primary, secondary, tertiary, motorway and trunk. The total reconstruction cost of roads exposed to landslide phenomena is
approximately 6.22 billion USD. The highest value of roads exposure belongs to a cell of the Jayl District (Chuy Region) in
the Kyrgyz Republic crossing by the EM-02 highway; the mean value is 110,240.00 USD per cell and about 0.40% of the
study area reports a value of exposure greater than 0.
The total reconstruction costs of different road classes exposed to landslides are reported in Table 2. It is worth noting that,
according to the spatial analysis, no motorway is directly affected by landslides phenomena and therefore the total motorway
exposed length is 0. Road reconstruction costs are proportional to the relevance of the road type (I.e. higher for trunk,
motorways and highways and lower for secondary and tertiary roads), but the total exposure of tertiary roads is nonetheless
higher than the one of primary and secondary roads because landslide-prone, mountainous areas are mostly covered by
tertiary roads.

| Typology | Exposure value | |
|---|---|---|
| Primary roads | Maximum | 340 thousand USD |
| | Mean | 129 thousand USD |
| | Sum | 851 million USD |
| Secondary roads | Maximum | 200 thousand USD |
| | Mean | 76 thousand USD |
| | Sum | 766 million USD |
| Tertiary roads | Maximum | 96,140.00 USD |
| | Mean | 36 thousand USD |
| | Sum | 1 billion USD |
| Trunk | Maximum | 800 thousand USD |
| | Mean | 303 thousand USD |
| | Sum | 3.6 billion USD |

**Table 2. Total reconstruction cost of each considered road class exposed to landslides in Central Asia.**




The spatial distribution of railways exposure is reported in Fig.4D, equally to roads exposure the total railways exposure has
been obtained summing the one of conventional railways with the high-speed one. Railways exposure reaches a maximum
value of 920 thousand USD, located in a cell of the Pop District of the Namangan Region in Uzbekistan and it is related to a
segment of the high-speed railway connecting the city of Tashkent with Andijan. The mean value is 344,000.00 USD per cell
and only the 0.03% of the cells are covered by a railway segment, highlighting that most of these linear infrastructures are
located in areas excluded from our grid analysis since are flat zones. The total exposed value of railways is about 1.23 billion
USD. In detail, the 98% of the total railways exposure is due to the high-speed one; the mean value of exposure of high-
speed railways is 349,425.00 USD per cell and the maximum value is the same of the total exposure. Contrarily,
conventional railways show a maximum value of 518,850.00 USD and the mean one is about 193,000.00 USD per cell. The
obtained results showed that railways exposure is greater than the one of roads and buildings, which is justifiable by their
high construction cost.




**Fig.4 Exposure maps of involved elements at risk. Panel A: population exposure; Panel B: building exposure; Panel C: road**
**exposure; Panel D: railway exposure.**





### 4.3 Landslide risk

Landslide risk analysis has been performed separately for each type of element at risk. Subsequently, the monetary value associated with different asset types was combined into a total risk map.

The specific risk of population is reported in Fig.5A and it ranges from 0 to 227 inhabitants. The maximum number of lives at risk is located in a cell of the city of Dushanbe in Tajikistan with a landslide hazard equal to 0.63 and a population exposure to 358.98 inhabitants, which corresponds to a density of 8974.5 inhabitants per km$^2$. The number of total lives at risk in Central-Asia is about 433,000 and the mean number is 3 inhabitants per cell. Equally to the specific risk of buildings, the population risk shows a very low mean number of lives at risk and it is surely related to the low percentage of cells (1.04% of grid analysis) where the number of lives at risk is greater than 0.

Fig. 5B shows the spatial distribution of landslide risk for buildings, which reaches a maximum value of 469,160.00 USD in a cell of the city of Almaty in Kazakhstan. This cell reports a landslide hazard value of 0.46 and a buildings exposure approximately to 1.02 million USD. The total risk associated with buildings in Central-Asia is about 186 million USD and the mean value is 8430.00 USD per cell. This value is relatively low when compared to the total exposed value of buildings in Central Asia. This is because the majority of buildings are located in areas where landslide hazard is very close or equal to 0. In fact, only the 0.77% of landslide-prone cells contain buildings, while most buildings in Central Asia are located in flat areas or in ones less prone to trigger landslides. However, specific landslide scenarios can still cause relevant losses at sub-national scale and should be analysed in detail with specific methods.

Specific landslide risk of roads is reported in Fig.5C, ranging from about 799,000.00 USD located in a cell of the Ohangaron District, region of Tashkent in Uzbekistan. This specific cell has a landslide hazard equal to 1 (very high probability of landslides occurrence); therefore, risk is equal to exposure. In this cell, exposure is high due to the presence of a segment of the A373 highway, connecting Osh (Kyrgyz Republic) and Tashkent (Uzbekistan) cities. The total landslides finantial risk associated with roads in Central-Asia is 3.02 billion USD and the mean value is about 58,000.00 USD per cell.

Regarding railways risk, its spatial distribution is showed in Fig.5D. Financial risk associated with railways ranges from 0 to 843,493.00 USD. Similarly, to roads risk the maximum value is located in the Ohangaron District, but in a different cell showing the following parameters: landslide hazard equal to 0.92 and railways exposure to 916,840.00 USD represented by the presence of a segment of high-speed railways. The obtained results report a mean value of 128,911.00 USD per cell and a total risk equal to 382 million USD. In general, for all exposed assets are located in few cells in the considered spatial domain. Besides, contrary to risk associated with building, the one for railways shows a high mean value considering that the cells covered by a railway segment are only the 0.03% of the grid analysis.

Therefore, our outcomes reveal that roads and railways are the element at risk that can be subjected to major losses respect to buildings, despite their minor covered area in the grid analysis. This is certainly due to the fact that railways and roads are built in areas more prone to trigger landslides respect buildings, that are mostly located in zones with landslide hazard very low or in flat areas.



**Fig.5 Landslide risk maps expressing the potential losses in terms of lives and economic damages for each involved element at risk. Panel A: population risk; Panel B: building risk; Panel C: road risk; Panel D: railways risk.**

Finally, the total risk expressed by the sum of the specific risk of buildings, roads and railways is showed in Fig.6. The maximum one is about 1.03 million USD. The highest landslide risk value is located in the same cell reporting the highest landslide risk of roads (Tashkent region – Uzbekistan). This cell shows the following parameters: landslide hazard equal to 1, building risk is 0, roads risk is about 799,000.00 USD and railways risk equal to 231,000.00 USD. The obtained results highlight that the total expected losses in Central-Asia are about 3.59 billion USD and a mean risk value of 23,401 USD per cell corresponding to 0.6 million USD/km$^2$; while the percentage of grid analysis with a landslide risk greater than 0 is approximately 1.10%, which are mostly located along the Tien-Shan chain or in areas at its foot. Inspecting the first ten cell with the highest risk values, we discovered that they are mainly located in the Ohangaron District (Uzbekistan) and the mean landslide hazard of these is 0.93. Besides, an already highlighted trend has been shown: the presence of specific exposed assets (railways) plays a relevant role in concurring to the total landslide risk in the region. In detail, these cells reported a





mean railways risk about of 587,000.00 USD per cell, which is greater than respective of buildings and roads, which are
often equal to 0.



**Fig.6 Total landslide risk map for Central -Asia. Panel A shows the distribution of potential economic losses across the whole study**
**area. Panel B shows a detail of the above map over the area covered by Tajikistan and Kyrgyz Republic.**






Fig. 7 shows the total landslide risk in Central-Asia aggregated within each district. The findings reveal that the district with
the highest possible losses is the Ayni District in Tajikistan with a total value of about 80 million USD (Fig. 8) and a
maximum one of 503,000.00 USD. The selected district is covered by the Tien-Shan chain and its landslide hazard values
range from 0.37 to 1, with a mean value of 0.55, revealing that the area is very prone to trigger landslides and to suffer
possible damages to structures and to loss of lives. Besides, the aggregation of landslide risk values at district level reveals
that the majority of these administrative units with high-risk values are mainly located in Tajikistan and Kyrgyz Republic,
which are the countries most affected by landslides and damages related to them in Central-Asia. Nevertheless, even
districts of other countries show high values of risk, for instance the Ohangaron District located in the region of Tashkent in
Uzbekistan is among the first ten districts with the highest total landslide risk (Fig. 8 A).
The obtained outcomes aggregated to the national-level further confirm our previous considerations about the landslide risk
distribution in Central-Asia and they show that landslide risk is mainly contributed by the one regarding roads, which ranges
from a minimum of 21 million USD in Turkmenistan to a maximum value of 682 million USD in Tajikistan (Fig. 8 B). In
detail, the risk component related to roads represents the 50% of the total risk at least (exception for Kazakhstan). This fact is
mainly due to the covered area of these infrastructures within the risk grid analysis, which is greater than the one related to
the other analysed elements at risk. Kyrgyz Republic shows the highest expected economic losses related to railways, with a
value of 324 million USD, nevertheless Uzbekistan is the country where railways risk more contributes to the total one with
a percentage of 42%. Finally, Kazakhstan reports the highest value of total buildings risk (33 million USD) across the
country in Central-Asia. Moreover, the aggregation at national level demonstrates that buildings component is always the
one characterized by the least weight within the risk analysis, this is because buildings are mainly located in areas where
landslide hazard is equal or close to zero or in alluvial plain, which are filtered off from our grid analysis.





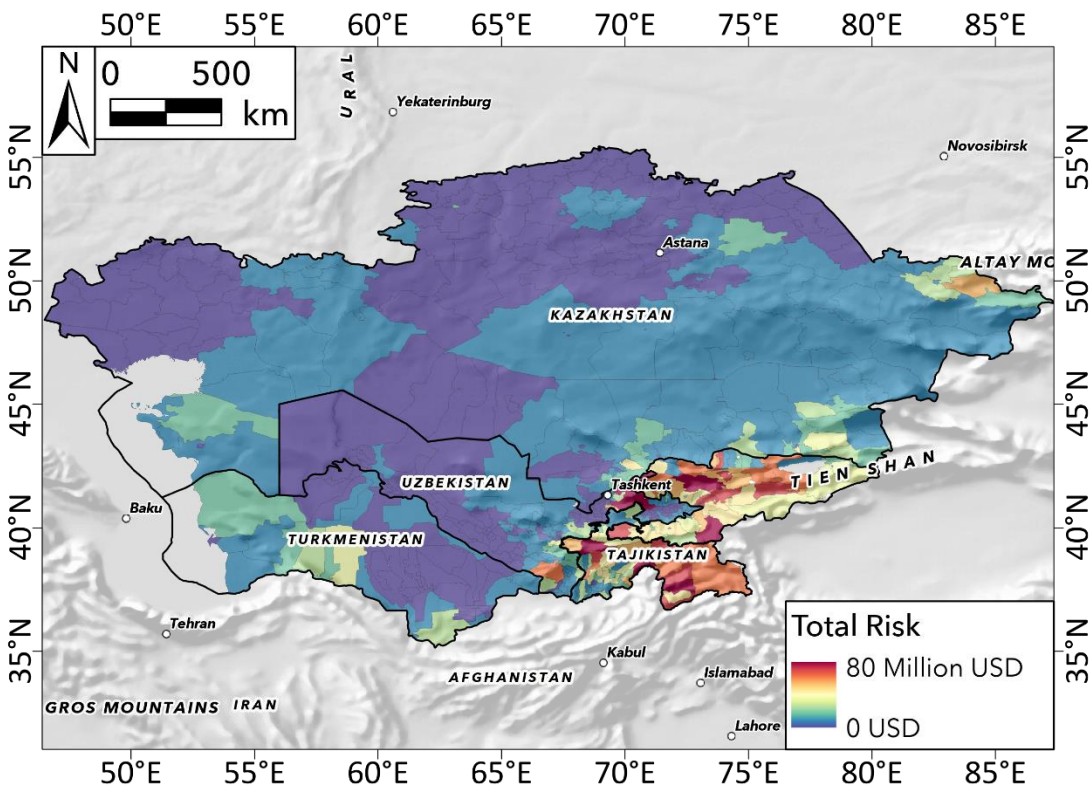

Fig.7 Total landslide risk map at district level in Central-Asia.


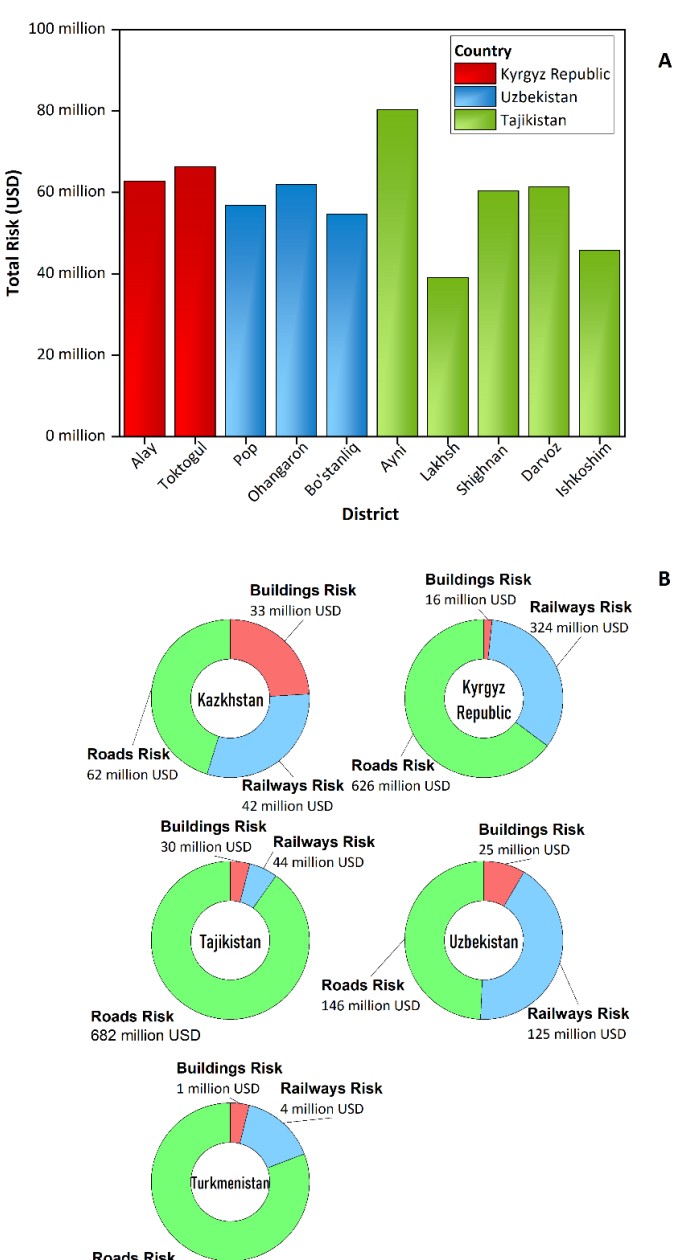


**Fig.8 Histogram of the ten districts with highest landslide risk in Central-Asia (Panel A). Landslide risk aggregated at national**

**level (Panel B).**





## 4.7 Considerations and future perspectives

In this study we performed a detailed quantitative analysis of landslide risk for the whole Central-Asia, which represents an advance step in the framework of risk analysis for very broad areas. The analysis was carried out using a 200 m spatial resolution and it was focused on the possible losses in terms of human lives (societal risk) and damages to human properties and infrastructures (financial risk). Regarding the economic losses, the risk analysis revealed that roads and railways are the elements that could be subjected to major damages at regional level due to their exposure and covered area instead of buildings, which are mainly located in flat areas. However, it should be noted that our study represents an attempt to estimate risk at regional scale and therefore some approximations within our workflow were adopted, such as the hazard or the vulnerability assessment. Nevertheless, outcomes of a small-scale analysis can be a useful tool for every developing country to get a preliminary outlook on the spatial distribution of possible losses and evaluate how cautionary are the administrative areas in planning its development. Furthermore, the performed analysis and highlighted approximations provide some general insights into which future developments could be focused. These could be certainly centred in evaluating in detail certain situations at sub-regional level (i.e. a downscaling phase) improving a time dimension in the landslide hazard framework and analysing the vulnerability of exposed elements in relation to possible impacts with these phenomena.

## 5 Conclusion

Landslides are a worldwide hazard, especially for developing countries due to the increasing of their urban development, population growth and drastic land use changes. The combination of these factors certainly influences the exposure in suffering social and economic damages related to landslides. Therefore, a quantitative risk analysis represents a useful tool to reduce possible consequences to human lives and properties due to landslides. In this research, we performed a quantitative risk analysis for the whole Central -Asia adopting a 200 m spatial resolution; landslide risk was analysed in terms of expected losses for population, human properties and infrastructure (buildings, roads and railways). The results showed that linear infrastructure are the exposed elements that could suffer the highest losses due to their location in areas very prone to trigger landslides. Furthermore, the findings highlight that the total expected losses in Central-Asia are about 3.59 billion USD and a mean risk value of about 0.6 million USD/ km $^2$.

Our study represents a significant advancement in the framework of risk analysis for extremely broad areas, however future development can be implemented into a downscaling phase in which evaluate some situations at sub-regional level improving the hazard and vulnerability assessment.



**Author contribution**

FC has conceived the research, written the manuscript, run the analyses. CS has contributed to the exposure assessment and to the revision of the manuscript. WF has contributed to the revision of the research. VT has conceived the research, supervised the work and revised the manuscript.

**Competing interest**

The contact author has declared that none of the authors has any competing interests.

**Acknowledgments**

This work was developed within World Bank-funded project "Strengthening Financial Resilience and Accelerating Risk Reduction in Central Asia" (SFRARR), in collaboration with the European Union, and the GFDRR (Global Facility for Disaster Reduction and Recovery), with the goal of improving financial resilience and risk-informed investment planning in the central Asian countries (Kazakhstan, Kyrgyz Republic, Tajikistan, Turkmenistan and Uzbekistan).

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
