# Peer review of "Regional-scale landslide risk assessment in Central-Asia"

_Natural Hazards and Earth System Sciences, 2023_

## Author Response (AR1)

**REVIEWER 1**

*This study conducted a quantitative risk analysis for landslides in Central Asia, covering an area of 4,000,000 square kilometers. The aim was to assess the potential impacts of landslides in terms of exposed population and expected economic losses to buildings, roads, and railways. The study highlighted the high susceptibility of developing countries in the region due to urban development, population growth, and land use changes. I agree that the findings could help decision-makers prioritize areas for intervention, allocate resources effectively, and enhance community resilience to landslides. The manuscript is well written and organized. Some minor concerns and suggestions:*

- *A down-scaling phase from the regional scale to the local scale could be implemented. Working at such a coarse scale, the analysis is heavily influenced by input data and the approximations made in the risk analysis.*
- *What about transitioning from regular grid cells to geomorphologically based mapping units, such as slope units, by replacing the regular grid partitioning with a delineation based on the characteristics of the terrain, such as slope, aspect, and topographic features.*
- *Figure 4 and 6 have legend and north in different positions with respect to the rest of the figures.*
- *Table 1. A more descriptive caption would be appreciated.*
- *I'd add a comparison with existing literature.*
- *how did the data scarcity affect the analysis? (add some discussion)*

Dear reviewer,

Thank you for your useful comments and feedback on our research. With this response, we would like to address the points you raised.

Regarding the first point, as stated in Section 4.7 "Considerations and future perspectives", a down-scaling phase could be implemented in the future. This latter would undoubtedly be a topic of future studies in which our results would be validated and compared using higher-resolution data. In addition, we did not include a down-scaling phase in the proposed work because we were not able to collect useful data to assess the hazard in its completeness and the vulnerability of the exposed elements.

Considering the second point, we deliberately chose to use regular grids instead of geomorphological units, as numerous tests have shown that they are the optimal mapping unit to extract the most heterogeneous information from the input data. Considering your feedback, the use of geomorphological units might be a reasonable mapping unit to adopt in the above-mentioned down-scaling phase.

Regarding Figures 4 and 6, we would like to thank you for your comment, we harmonized them with the rest of the figures. Please, see the revised version of the manuscript.

Regarding Table 1, we thank you for the suggestion and we added a more detailed caption in the manuscript.

Concerning the point on the comparison with existing literature we decided to improve our introduction section by adding a brief paragraph at **Line 84:**

" The predominant factor contributing to the lack of studies focused on landslide risk at small-scale is primarily attributed to challenges associated with accessing data pertaining to each element within the risk equation. However, recent advancements in acquiring global digital data opened up the potential to bypass the drawbacks of landslide risk analysis and generate preliminary analyses for broad geographic areas that were previously beyond reach."

As for the final point, we opted to insert further considerations into section 4.7 "Considerations and future perspectives" at **line 402:**

"Notably, data scarcity in landslide studies can significantly hinder the accurate evaluation of the risk posed by these phenomena, potentially putting communities at greater risk (Uzielli et al., 2015a; Dragićević et al., 2015; Jacobs et al., 2018). Furthermore, limited data can impede the development of effective early warning systems (Peres and Cancelliere, 2021; Marin et al., 2021; Lindsay et al., 2022). Indeed, without access to useful data needed to estimate the components of landslide risk equation (e.g landslide hazard in its completeness or vulnerability of exposed elements), it becomes challenging to produce reliable products (Biçer and Ercanoglu, 2020)."

REFERENCES:

Uzielli, M., Catani, F., Tofani, V., & Casagli, N. (2015). Risk analysis for the Ancona landslide—I: characterization of landslide kinematics. Landslides, 12, 69-82.

Dragićević, S., Lai, T., & Balram, S. (2015). GIS-based multicriteria evaluation with multiscale analysis to characterize urban landslide susceptibility in data-scarce environments. Habitat international, 45, 114-125

Jacobs, L., Dewitte, O., Poesen, J., Sekajugo, J., Nobile, A., Rossi, M., ... & Kervyn, M. (2018). Field-based landslide susceptibility assessment in a data-scarce environment: the populated areas of the Rwenzori Mountains. Natural Hazards and Earth System Sciences, 18(1), 105-124.

Biçer, Ç. T., & Ercanoglu, M. (2020). A semi-quantitative landslide risk assessment of central Kahramanmaraş City in the Eastern Mediterranean region of Turkey. Arabian Journal of Geosciences, 13, 1-26.

Peres, D. J., & Cancelliere, A. (2021). Comparing methods for determining landslide early warning thresholds: potential use of non-triggering rainfall for locations with scarce landslide data availability. Landslides, 18(9), 3135-3147.

Marin, R. J., Velásquez, M. F., García, E. F., Alvioli, M., & Aristizábal, E. (2021). Assessing two methods of defining rainfall intensity and duration thresholds for shallow landslides in data-scarce catchments of the Colombian Andean Mountains. Catena, 206, 105563.

Lindsay, E., Frauenfelder, R., Rüther, D., Nava, L., Rubensdotter, L., Strout, J., & Nordal, S. (2022). Multi-temporal satellite image composites in google earth engine for improved landslide visibility: A case study of a glacial landscape. Remote Sensing, 14(10), 2301.

**REVIEWER 2**

*The topic is of interest because it analyses the risk of landslides on a regional scale, as far as Central Asia is concerned. The results are relevant for policy makers in order to take appropriate measures to reduce landslide risk, especially the one which affects the linear structures. As flat areas are not prone to landslide triggering, it may be better to also present detailed images of landslide affected areas in Figures 3, 4 and 5. At one point the issue of landslide induced risk is addressed. Since the induced risk is a matter of anthropogenic influence, and the assessment here considers landslide risk as a whole, I do not think that this sentence "The landslide-induced risk has been calculated in terms of exposed population and expected economic losses for buildings and linear infrastructure (roads and railways)" can be taken into account, since any type of risk is calculated in these terms, whether anthropogenic or natural. In Table 1 the input data in the caption should be described. Row 183 - 200 enter m, row 187 - 100 enter m, row 188 - 200 enter m.*

Dear reviewer,

We thank you for your comments and interest in our research.

Regarding the inclusion of detailed images of landslide-affected areas, we are not able to add them since we didn't do any field investigation. Furthermore, the use of web images requires the consensus and the copyright of authors. We are sorry about that, it would have been helpful and for this reason, we would like to thank you for the valuable suggestion.

Concerning the sentence " *The landslide-induced risk has been calculated in terms of exposed population and expected economic losses for buildings and linear infrastructure (roads and railways)* " , it underscores that the risk assessment conducted was grounded in quantitative analysis. Specifically, it quantified the risk to inhabitants per mapping unit and defined the risk to buildings and linear infrastructures in terms of expected economic losses (measured in USD) in the event that a landslide affects them. Therefore, this sentence holds significance as it introduces the audience to the types of outcomes derived from the assessment.

The caption of Table 1 has been improved. Many thanks for your suggestion.

**REVIEWER 3**

*The paper presents an interesting and potentially publishable piece of research that would be of interest to both the wider academic research community as well as stakeholder in the industry. However, I recommend a major revision in terms of contents and editing.*

*Please refer to the comments below.*

- *The main shortcoming is a lack of clear research framework that should contain a clearly set main research goal and objectives, research methodology and research hypothesis.*
- *Further explanations of the potential advantages and weaknesses of this study are needed.*
- *The paper may be improved by providing tables about the data collection was added to help readers understand.*
- *Further explanation is needed for the differentiation and superiority of the study.*
- *In the discussion session, please explain in detail examples where the results of this study can be applied to the actual field.*
- *The conclusion should be more clearly explained with additional discussion and the value of findings/analysis*

Dear reviewer,

We appreciate your interest in our study, and we would like to thank you for your valuable insights. With this answer, we would like to address the raised points by you.

In light of the first and second point, we rectified this shortcoming by improving our "Introduction" section with the addition of a paragraph (**at Line 88)** aimed at defining our research goal, advantages and weaknesses:

" Based upon these developments, the main objective of this research is to undertake an exhaustive landslide risk assessment in quantitative terms, focusing on a geographically broad area encompassing the entirety of Central Asia (about 4,000,000 km2). Despite historical evidence of substantial damage caused by landslides within this region, it is notable that, to date, a comprehensive landslide risk assessment at a regional scale remains conspicuously absent in the scientific literature. The motivation for production is based on the expected increase in landslide-related risk in Central Asia due to several factors, including but not limited to increased urbanization, population growth, and dramatic land use change. These evolving dynamics will drive up the risk of landslide-related losses in the region. This work is primarily concerned with evaluating and disseminating the first regional-scale landslide risk assessment for Central Asia. This comprehensive assessment will facilitate approaches and decisions for mitigation strategies at the regional scale. The focus of the proposed analysis is to quantify landslide-related risk in terms of two distinct facets: the population exposed to landslides and the expected economic losses associated with damage to buildings and linear infrastructure, particularly roads and railways. Given the vast extent of the selected region as the subject of our study, we acknowledge that certain approximations should inevitably be integrated within the framework of our analysis. In light of these approximations, there is certainly a degree of overestimation. Indeed, we assume that in the event of a landslide, all elements located in a mapping unit would suffer irreparable damage, and this concept boils down to considering their maximum degree of vulnerability. The ultimate goal of this research is to

identify the areas in Central-Asia where the propensity for high losses from landslides is most pronounced. The insights that this analysis can provide are intended to be a valuable resource in facilitating effective mitigation measures and land-planning policies."

Regarding the point on providing tables on the data collection, we would like to stress the presence of Table 1, where the references and main characteristics of input data are reported. Furthermore, we improved the caption of this Table thanks to suggestions provided by Reviewer 1 and Reviewer 2.

Concerning the third and fourth points, we acknowledge the need for further explanations. Therefore, we opted to improve section 4.7 "Considerations and future perspectives" taking into account the suggestions provided by Reviewer 1 as well. Here, is the new version of the above-mentioned section:

" In the context of this research, we undertook a quantitative assessment of landslide risk in Central-Asia. Our analytical framework involved a spatial resolution of 200 m and a focus on the quantification of potential losses, encompassing both human lives and economical losses associated with the damage to human settlements and linear infrastructures. The findings of this regional-scale landslide risk assessment constitute an innovative step forward, as such comprehensive assessments for vast geographic regions have historically been scarce in the scientific literature. Despite this, we would like to recall once more the inherent limitations mainly stemming from data scarcity, which make arduous to evaluate some landslide risk components, as the assessment of the temporal and areal probability of landslide occurrence. Notably, data scarcity in landslide studies can significantly hinder the accurate evaluation of the risk posed by these phenomena, potentially putting communities at greater risk (Uzielli et al.,2015; Dragićević et al., 2015; Jacobs et al., 2018). Furthermore, limited data can impede the development of effective early warning systems (Peres and Cancelliere, 2021; Marin et al., 2021; Lindsay et al., 2022). Indeed, without access to useful data needed to estimate the components of landslide risk equation (e.g landslide hazard in its completeness or vulnerability of exposed elements), it becomes challenging to produce reliable products (Biçer and Ercanoglu, 2020). Moreover, the adoption of a 200-m spatial resolution may obscure the socio-economic heterogeneities across Central Asia, thereby rendering our risk estimates as generalized approximations. However, it should be noted that findings resulting from a small-scale analysis can represent a valuable initial resource for any developing country (Stanley and Kirschbaum, 2017; Sim et al., 2022). These analyses provide a preliminary outlook on the spatial distribution of potential losses and offer insights into the degree of prudence required within administrative regions when formulating spatial planning strategies. In a rising context, where accurate data for in-depth assessments may be limited, small-scale analyses can play a fundamental role by delineating spatial patterns associated with potential losses, which can help policymakers and stakeholders in their efforts to produce a resilient suistanable development framework. Undoubtedly, the inherent limitations necessitate further investigation and refinement to attain more detailed findings. In this perspective, future developments should be focused on in depth-studies at the sub-national level (e.g. a down-scaling phase) with the objective of evaluating in detail all the risk components."